Characterization of a biofilm-forming Shigella flexneri phenotype due to deficiency in Hep biosynthesis

Xu Dan dan.xu@xjtu.edu.cn 1
Zhang Wei 1
Zhang Bing 1
Liao Chongbing 1 2
Shao Yongping yongping.shao@xjtu.edu.cn 1 2
1 Key Laboratory of Biomedical Information Engineering of the Ministry of Education, School of Life Science and Technology, Xi’an Jiaotong University , Xi’an , Shaanxi , China
2 Center for Translational Medicine, Frontier Institute of Science and Technology, Xi’an Jiaotong University , Xi’an , Shaanxi , China
Diao Jiajie
Electronic publication date: 2016 Jul 14
Publication date: 2016
Volume: 4
Electronic Location ID: e2178
Received 2016 May 5; Accepted 2016 Jun 5
Copyright: ©2016 Xu et al.
Copyright year: 2016
Copyright holder: Xu et al.
License: This is an open access article distributed under the terms of the Creative Commons Attribution License, which permits unrestricted use, distribution, reproduction and adaptation in any medium and for any purpose provided that it is properly attributed. For attribution, the original author(s), title, publication source (PeerJ) and either DOI or URL of the article must be cited.
License URL: https://creativecommons.org/licenses/by/4.0/

Keywords: Shigella flexneri, Biofilm formation, Enhanced adhesion

Funding: Natural Science Foundation of China 31401211 China Postdoctoral Science Foundation 2014M552428 2015T81014 This work was supported by grants from the Natural Science Foundation of China (31401211 to DX), China Postdoctoral Science Foundation (2014M552428 and 2015T81014 to DX). The funders had no role in study design, data collection and analysis, decision to publish, or preparation of the manuscript.

==============================
Deficiency in biosynthesis of inner core of lipopolysaccharide (LPS) rendered a characteristic biofilm-forming phenotype in E. coli. The pathological implications of this new phenotype in Shigella flexneri, a highly contagious enteric Gram-negative bacteria that is closely related to E. coli, were investigated in this study. The ΔrfaC (also referred as waaC) mutant, with incomplete inner core of LPS due to deficiency in Hep biosynthesis, was characteristic of strong biofilm formation ability and exhibited much more pronounced adhesiveness and invasiveness to human epithelial cells than the parental strain and other LPS mutants, which also showed distinct pattern of F-actin recruitment. Failure to cause keratoconjunctivitis and colonize in the intestine in guinea pigs revealed that the fitness gain on host adhesion resulted from biofilm formation is not sufficient to offset the loss of fitness on survivability caused by LPS deletion. Our study suggests a clear positive relationship between increased surface hydrophobicity and adhesiveness of Shigella flexneri, which should be put into consideration of virulence of Shigella, especially when therapeutic strategy targeting the core oligosaccharide (OS) is considered an alternative to deal with bacterial antibiotics-resistance.

Introduction

Shigella flexneri is a highly contagious facultative intracellular pathogen causing acute inflammatory enteritis in human. Following infection of epithelial cells and macrophages, S. flexneri escapes from the phagocytic vacuoles via Ipa-mediated vacuole membrane lysis and enters the cytoplasm, where active proliferation takes place (Philpott, Edgeworth & Sansonetti, 2000; Schroeder & Hilbi, 2008). Bacteria then spread intra- and intercellularly by recruiting the cellular actin polymerization machinery with the help of the intra/intercellular spread factor, IcsA (Lett et al., 1989; Suzuki & Sasakawa, 2001). While the invasion process and immune evasion of S. flexneri have been extensively studied, the early adhesion process has not been adequately understood since Shigella flexneri lacks general adhesion machinery and exhibits relatively poor adhesiveness compared with other pathogenic enterobacteria (Brotcke Zumsteg et al., 2014; Carayol & Tran Van Nhieu, 2013; Phalipon & Sansonetti, 2007; Pizarro-Cerdá & Cossart, 2006).

LPS is a glycolipid located in the outer membrane of Gram-negative bacteria. It is composed of three covalently linked domains: lipid A, which is embedded in the outer membrane; the oligosaccharide core including inner and outer parts; and repeats of the O-polysaccharide or O-antigen, which cover the bacterial surface. As an essential pathogenic component, LPS of gram-negative bacteria triggers strong immune responses, which are directly related to the adverse clinical outcomes (Alexander & Rietschel, 2001; Wang & Quinn, 2010). In contrast to the highly variable and antigenic O-antigen portion, the core oligosaccharide (OS), especially the inner (lipid A-proximal) core, composed of two 3-deoxy-Dmanno-oct-2-ulosonic acids (Kdo) and three L-glycero-D-mannoheptose (Hep), called HepI, HepII, and HepIII, is conserved across E. coli, Shigella and Salmonella and possesses limited structural variation. Therefore, targeting the core OS for general therapeutic application has been considered as an alternative strategy against antibiotics-resistant Gram-negative bacterial infection (Desroy et al., 2009; Di Padova et al., 1993; Moreau et al., 2008). Although inhibition of Kdo biosynthesis is usually lethal to bacteria, a defect in Hep biosynthesis results in a viable bacterial cell with a characteristic “deep rough” phenotype (Grizot et al., 2006; Klena et al., 2005).

Biofilm, a microorganism community formed on an environmental surface and by cell aggregation has been implicated in around 80% of microbial infections in vivo (Hall-Stoodley, Costerton & Stoodley, 2004) (Davies, 2003). Nevertheless, the relevance of biofilm formation to Shigella virulence has not been thoroughly interrogated. Recent studies have shown that LPS composition regulates biofilm formation besides the previously reported association with variations in salt concentration, starvation or changes in pH (Ellafi et al., 2012). For example, deficiency in Hep synthesis in E. coli resulted in dramatic biofilm formation on abiotic surface, which is caused by enhanced hydrophobicity of the bacteria surface due to the loss of oligosaccharide (Nakao et al., 2012). Although Shigella LPS has been extensively studied through deletion of a serious of genes involved in LPS synthesis, disruption of Hep biosynthesis has never been reported in Shigella so far (Hong & Payne, 1997; Martini et al., 2011; Sandlin et al., 1995).

In this study, we characterized biofilm forming potentials and pathological behaviors of various LPS-truncated Shigella flexneri strains. Analysis of the LPS mutants revealed that the autoaggregation and biofilm forming capacity of Shigella negatively correlated with the LPS chain length in general. The deep-rough LPS mutant ΔrfaC, characteristic of strong biofilm formation abilities, exhibited much more adhesive and invasive than the parental strain and other LPS mutants, albeit with undermined fitness. Immunofluorescence analysis revealed that the ΔrfaC strain exhibited distinct patterns of IcsA distribution and F-actin recruitment. Finally, using the guinea pig keratoconjunctivitis model, we showed that the biofilm-forming Shigella strain failed to colonize in vivo, indicating that the fitness gain on host adhesion resulted from biofilm formation is not sufficient to offset the loss of fitness on survivability caused by LPS deletion.

Materials & Methods

Bacterial strains and plasmids

Bacterial strains and plasmids used in this study are listed in Table 1. Shigella strains were cultured aerobically at 37°C in Tryptic Soy (TS) broth (Aoboxing, Beijing, China) or on TS agar plates with 0.1% Congo Red. Antibiotics (Sigma) were used as follows: ampicillin 100 µg/ml; kanamycin 100 µg/ml.

Table 1 Strains and plasmids in this study.

Strains	Relevant genetype, phenotype and description	Reference	
Shigella flexneri			
Sf301	Shigella flexneri 2a strain	(Jin et al., 2002)	
301ΔicsA	Sf301 ΔicsA (+51 to +3255): KRC	This work	
301Δwzy	Sf301 Δwzy (+26 to +1075): KRC	This work	
301ΔwaaL	Sf301 ΔwaaL (+6 to +1155): KRC	This work	
301ΔrfaC	Sf301 ΔrfaC (+33 to +832): KRC	This work	
Plasmids			
pKD4	Template plasmid for λ Red recombination system	(Datsenko & Wanner, 2000)	
pKD46	λ Red recombinase expression plasmid	(Datsenko & Wanner, 2000)	
pCP20	λ Red FLP-recombinase expression plasmid	(Datsenko & Wanner, 2000)	

Strain construction

Bacterial gene knockout was performed using the λ Red recombination system (Datsenko & Wanner, 2000). Briefly, bacterial cells transformed with pKD46 were grown in the presence of L-arabinose to induce the expression of the lambda Red recombinase. A linear PCR product, amplified using the primers listed in Table 2, containing a kanamycin-resistance cassette (KRC) flanked by FLP and 50 bp of the 5′- and 3′-end homologous sequences of the target gene was electroporated into the bacterial cells and kanamycin was used to select the transformants. The plasmid pKD46 was eliminated by incubation at 37°C. To cure the kanamycin maker, pCP20 was introduced into kanamycin-resistant cells to elicit the recombination of flanking FLP sequences at both ends of the kanamycin cassettes. PCR screening for cured colonies were performed using specific primers listed in Table 2.

Table 2 Oligonucleotides used in this study.

Name	Sequence	Target	
icsARedF	ATGAATCAAATTCACAAATTTTTTTGTAATATGACCCAAT GTTCACAGGGGTGTAGGCTGGAGCTGCTTC	KRC targeting icsA	
icsARedR	AAGGTATATTTCACACCCAAAATACCTTGGGTGTCTCTGT AACTGTTATTATGGGAATTAGCCATGGTCC	KRC targeting icsA	
icsAF	GACCCAATGTTCACAGGG	icsA	
icsAR	TGGGTGTCTCTGTAACTG	icsA	
wzyRedF	ATAACTTCCCTATTTTTAACATCCTTTATTTTGCTCCAGAA GTGAGGTTAGTGTAGGCTGGAGCTGCTTC	KRC targeting wzy	
wzyRedR	ATAACATTTTTATGTATTGAACTGATTATTGGTGGTGGTGG AAGATTACTATGGGAATTAGCCATGGTCC	KRC targeting wzy	
wzyF	TTGCTCCAGAAGTGAGG	wzy	
wzyR	GTGGTGGTGGAAGATTAC	wzy	
waaLRedF	CTCAACATTATTTTTCTCTCTCGAGAAAAAAAACTGGATAGC GTACTGGAGTGTAGGCTGGAGCTGCTTC	KRC targeting waaL	
waaLRedR	TTGTTTTTCATCGCTAATAATAAGCCGGCGTAAACGCCTAAT AAATTTGGATGGGAATTAGCCATGGTCC	KRC targeting waaL	
waaLF	ACTGGATAGCGTACTGG	waaL	
waaLR	TAAGCCGGCGTAAACGC	waaL	
rfaCRedF	CACTGATGCCCAGCAGGCAATCCCAGGGATTAAGTTTGACTG GGTGGTGGGTGTAGGCTGGAGCTGCTTC	KRC targeting rfaC	
rfaCRedR	AAGAGACATACTTGTAGAACGACACTCTACTTGATTCTTCCCA TACCCACATGGGAATTAGCCATGGTCC	KRC targeting rfaC	
rfaCF	GGGATTAAGTTTGACTGG	rfaC	
rfaCR	GTAGAACGACACTCTAC	rfaC	

LPS preparation and electrophoresis

LPS was prepared from Shigella strains as described by Hitchcock & Brown (1983). Bacteria grown overnight on TSA were resuspended in PBS to an OD600 of 0.8. The bacterial pellet from 1.5 ml of the suspension was then resuspended in 125 ml lysis buffer (0.1 M Tris/HCl pH 6.8, 2% SDS, 4% β-mercaptoethanol and 10%, v/v, glycerol) and boiled for 10 min. Proteinase K (50 mg, Sigma) was added and the mixture incubated for 1 h at 60°C. Samples were run on a Tris-Tricine gel) and visualized by silver staining (Hitchcock & Brown, 1983).

Autoaggregation assay

Overnight-cultured bacteria were harvested by centrifugation (10,000× g for 2 min) and two ml of whole cells standardized at OD600 = 1 after suspension in PBS were placed in a 14-ml polyethylene tube and incubated at 4°C under a static condition. The OD600 of the phase above the sediment by aggregation was recorded at different time points.

Biofilm formation assays

Biofilm formation by Shigella strains was assayed as previously described with some modifications. For biofilm analysis on polystyrene surface, 107 CFU of Shigella in 100 µl of TSB broth was inoculated into the wells of a 96-well flat-bottom polystyrene microtiter plate. The bacterial strains were grown at 37°C for 48 h under a static condition and the planktonic cells in liquid medium were discarded. The plate or tube was washed twice with distilled water and air-dried. Attached biofilms were stained with 0.1% crystal violet for 20 min. Then, the plates were rinsed twice with distilled water to remove excess stain and air-dried. In order to quantify the amount of biofilm on a 96-well plate, all stain associated with the attached biofilms was dissolved with 95% ethanol, then OD595 absorbance was measured using a microplate reader.

To prepare the biofilm sample for Confocal laser scanning microscopy (CLSM) analysis, GFP-expressing plasmid was electroporated into Shigella strains. 108 CFU of GFP-Shigella was added in 1 ml of TSB broth per well and incubated at 37°C sunder a static condition. Biofilms were grown on cover glass (φ = 14 mm) placed in 24-well polystyrene cell culture plates. After 48 h, the cover glass was rinsed twice with PBS to remove any planktonic cells. After washing, the cells were fixed in 3% paraformaldehyde/PBS and mounted with Anti-Fade solution (Invitrogen) containing DAPI onto glass slides. After the preparation, the samples were examined under the confocal laser scanning microscope ZEISS LSM 710 (Carl-Zeiss), and images were processed by LSM software ZEN (Carl-Zeiss, Oberkochen, Germany).

To prepare the biofilm sample for scanning electron microscopy (SEM), the biofilms grown in the cover glass dehydrated in graded ethanol, critical point dried with CO2 and coated with gold-palladium beads with a diameter of 15 nm. Samples were photographed using a Philips XL-30 scanning electron microscope at 20 kV.

In vitro adhesion and invasion assays and microscopy

One day before the assays, HeLa cells (ATCC CCL-2) were seeded into 24-well plates at a density of ∼105 cells per well. One hour before the infection, cell culture medium were changed into serum-free medium and ∼106 CFU Sf301 or its LPS mutants from mid-exponential phase was added to the cells together. Bacteria were centrifuged (2,000 rpm, 10 min, RT) onto HeLa cells (moi 10:1, or indicated moi) to synchronize the infection. For adhesion assay, after washing, the cells were lysed with distilled water and the CFU was enumerated after plating. For invasion and proliferation assays, bacteria/HeLa mixtures were incubated for 40 min after centrifugation and then washed, treated with gentamycin-containing (25 µg/ml) medium for another 1 h (invasion) or 4 h (proliferation) before lysed for plating. Adhesion was defined as the total number of HeLa cell-associated bacteria and is shown as the percentage of input. Invasion and proliferation was defined as the total number of intracellular bacteria in cells (extracellular bacteria was killed by gentamycin, a cell-impermeable antibiotic). Average results of three independent experiments are shown as mean ± SD.

For fluorescent microscopy of bacterial adhesion to HeLa cells, cells were plated onto glass coverslips and adhesion assays were performed as described using Sf301 harboring the GFPUV-expressing plasmid (moi 100:1). Cells were fixed in 3% paraformaldehyde/PBS at room temperature for 15 min, washed in PBS and mounted with Anti-Fade solution (Invitrogen) containing DAPI onto glass slides and visualized under Zeiss confocal microscope. For scanning electron microscopy (SEM), the cover glass was dehydrated in graded ethanol, critical point dried with CO2 and coated with gold-palladium beads with a diameter of 15 nm. Samples were photographed using a Philips XL-30 scanning electron microscope at 20 kV.

Lactate dehydrogenase (LDH) activity assay

The LDH assay was performed using the LDH cytotoxicity assay detection kit (Beyotime, China), according to the manufacturer’s instructions. The assay measures the conversion of a tetrazolium salt to a red formazan product, detectable by absorbance measurement at 490 nm. For this, a SpectraMax 190 Microplate Reader (Molecular Devices) was used.

Phalloidin staining assay

For phalloidin staining of F-actin in Shigella-infected HeLa cells, cells were plated onto glass coverslips and invasion assay was performed as described using Sf301 harboring the GFPUV-expressing plasmid (moi 100:1) for two hours. Cells were fixed in 3% paraformaldehyde/PBS at room temperature for 15 min, washed in PBS and permeabilized in 0.1% Triton-X100 in PBS for 1 min. F-actin were stained with 80nM TRITC-phalloidin (Yeasen, Shanghai, China) for 30 min. The coverslips was mounted with Anti-Fade solution (Invitrogen) containing DAPI onto glass slides and visualized under Zeiss confocal microscope.

Plaque assay

This was carried out according to Oaks, Wingfield & Formal (1985) using confluent HeLa cell monolayers. Briefly, HeLa cells to be used in the plaque assay were grown to 100% confluency in 6-well polystyrene plates (Corning) in DMEM supplemented with 10% FBS. One hour before the infection, cell culture medium were changed into fresh medium and Sf301 or its LPS mutants from mid-exponential phase (m.o.i 100:1) was added to the cells and subsequently incubated at 37°C for 90 min. During this adsorption or attachment phase, the plates were rocked every 30 min to assure uniform distribution of the plaque-forming bacteria. Next, an agarose overlay (5 ml) consisting of DMEM, 5%FBS, 25 µg of gentamicin per ml, and 0.5% agarose was added to each plate. The plates were incubated at 37°C in a humidified 5% CO2 and examined daily for up to 3 days for plaque formation. The agar was carefully removed three days later and the HeLa monolayers were stained by Giemsa staining kit (Yeasen, Shanghai, China) in order to visualize the plaques.

Sereny test

Female Hartley guinea pigs, aged 6–8 weeks, weighing 200–300 g, were inoculated with 107 CFU/eye of mid-log phase Sf301 and its LPS mutants via conjunctival route as described, with 3 animals in each group. The protocol has been approved by the Animal Research Ethical Committee of School of Life Science and Technology the Xi’an Jiaotong University (Approval No. 201411). Inoculated animals were observed and scored for 7 consecutive days for development of the conjunctivitis. Eyes were blindly scored by three individuals (DX, YPS, YC) on a scale of 0–3 defined as follows: grade 0 (no disease or mild irritation), grade 1 (mild conjunctivitis or late development and/or rapid clearing of symptoms), grade 2 (keratoconjunctivitis without purulence), grade 3 (fully developed keratoconjunctivitis with purulence).

Guinea pigs were anesthetized using chloral hydrate (10 mg/kg of body weight) before being inoculated via an intrarectal (i.r.) route with 108 CFU of Sf301 and its LPS mutants in 100 µl PBS, with 3 animals in each group. The protocol has been approved by the Animal Research Ethical Committee of the Xi’an Jiaotong University. Three pieces of feces of each inoculated animal were collected at the same time of the day for 18 days. Each piece of the feces were dispersed and 1 ml PBS. After centrifuge, 100 µl of the supernatant was plated onto MaConkey plates. Shigella bacteria were recognized as smooth, white colonies on MaConkey plates and further confirmed by PCR of spa33 gene.

Figure 1 LPS structures, growth and TEM analysis of the Shigella flexneri LPS mutants in this study.

(A) Illustrative drawing of LPS structures of Sf301, Δwzy, ΔwaaL and ΔrfaC strains is showed in the upper panel. Kdo: 3-deoxy-D-mannooct-2-ulosonic acid; Hep, L-glycero-D-manno-heptose phosphate; PEtN, O-phosphoryl-ethanolamine; Glc, D-glucose; Gla, d-galactose; GlcNAc, N-acetyl-D-glucosamine; Rha, L-rhamnose. The LPS of these strains was analyzed by silver staining of polyacrylamide gel after SDS-PAGE. (B) Growth of Sf301 and the LPS mutants. Each strain was grown in TSB broth under shaking conditions at 37°C. Absorbance at OD600 was measured at different time points. (C) Representative transmission electron microphotographs (TEM) of each strain are shown. Overnight culture of each strains were collected and stained by with 1.5% phosphotungstic acid for 90 s and examined under TEM. Arrows indicate the dark spots on the surface of ΔrfaC strain.

Statistical analysis

All data are presented as means ± SEMs. Statistical analysis was performed within the subgroups by using Student’s t-test for autoaggregation assays, biofilm formation assays, in vitro adhesion/invasion assays and LDH assays in this study. p values of less than 0.05 were regarded as statistically significant. Statistical analysis was performed with GraphPad Instat 3 software (GraphPad Software, Inc, La Jolla, Calif).

Results

Generation and characterization of the LPS mutants of Shigella flexneri

To generate Shigella LPS mutants of different chain length, we knocked out wzy, waaL and rfaC individually using the lambda red system (Fig. S1). As shown in Fig. 1A, the wzy mutant contains a complete core and one copy of the O-antigen; the waaL mutant has a complete core but no O-antigen; the rfaC mutant is deficient of Hep but retains the Kdo. SDS-PAGE analysis confirmed that the all mutant strains produced proper LPS variants as expected (Fig. 1A). The Δwzy and ΔwaaL strains formed WT-like colonies on Tryptone Soya Agar, while the ΔrfaC strain grew round and smooth colonies with smaller size. In accordance with this, the ΔrfaC mutant grew slightly slower in liquid tryptone soya broth medium than other strains (Fig. 1B). LPS truncation had no apparent impact on the morphology of individual bacteria cells as illustrated by transmission electron microscopy (TEM) (Fig. 1C).

Figure 2 Analysis of biofilm formation of LPS mutants of Sf301.

(A) Autoaggregation phenotype by LPS mutants. Each strain standardized at OD600 = 1.0 in PBS was used for autoaggregation assay. The value at OD600 after an18-hour incubation is shown as the mean ± SD of results from three independent experiments. Statistical analysis was performed using student t-test. *, P < 0.01 against autoaggregation level of strain Sf301. (B) Biofilm formation by Shigella flexneri LPS mutants in 96-well flat-bottom polystyrene microtiter plate when compared to the parental strain. The mean ± SD of results from three independent experiments are shown. Statistical analysis was performed using student t-test. *, P < 0.01, against biofilm formation level of strain Sf301. (C) Biofilms on glass cover slides of LPS mutants stained by DAPI. (D) CLSM and SEM analysis of biofilms formed by ΔrfaC strain. A section which has representative signals in the defined area is shown in the left column (X–Y). The overview of biofilms in the same area of each X–Y section is shown as 3D image in the middle column (3D). Biofilms under SEM are shown in the right column. (E) Influence of DNase I treatment on the biofilm formation of ΔrfaC strain. ΔrfaC strain was grown in presence of different concentrations of DNase I or in presence of pre-heated DNase I or without DNase I for 48 h under static conditions at 37°C. Statistical analysis was performed using student t-test. *, P < 0.01, against the biofilm formation by ΔrfaC strain without DNase I treatment. (F) biofilm formed by ΔrfaC strain under high-magnification (10000X) SEM. Arrows indicate the fibrous connections among bacteria.

Biofilm formation is enhanced in the rfaC-deleted Shigella strain

The aggregation tendency of LPS mutants was initially evaluated in PBS and the results showed that LPS truncation led to enhanced bacterial aggregation (Fig. 2A). Next, we examined the biofilm formation of the LPS mutants on the polystyrene surface under a static culture condition. While the WT strain had a poor ability to form biofilm, the LPS mutants showed improved potential to do so (Fig. 2B). Moreover, the ability to form biofilm appeared to be negatively correlated with the LPS length with the ΔrfaC strain (shortest LPS) being the most effective one (Fig. 2B). Fluorescence microscopy and scanning electron microscopy (SEM) further confirmed the enhanced biofilm formation by the ΔrfaC strain on glass surface (Figs. 2C–2D). To test the involvement of extracellular DNA in the biofilm formation by the ΔrfaC strain, DNase I was added when the bacteria were seeded into the polystyrene plates. Disruption of extracellular DNA significantly abolished the formation of biofilm without affecting the viability of the bacteria (Fig. 2E and Fig. S2). In addition, SEM analysis demonstrated that there were fibrous connections among bacterial cells in the biofilm (Fig. 2F).

Figure 3 Invasion and adhesion ability of Shigella flexneri LPS mutants in comparison with the parental Sf301 strain.

(A) Invasion and intercellular proliferation of Sf301 and its LPS mutants. (B–C) Initial adhesion of Sf301 and its LPS mutants at 30°C and 37°C to HeLa cells. Briefly, Bacteria were centrifuged onto HeLa cells (moi 10:1) to synchronize the infection. For adhesion assay, after washing, the cells were lysed with distilled water and the CFU was enumerated after plating. For invasion and proliferation assays, invasion was allowed to happen for 40 min after centrifuge, followed by washing and 1-hour (invasion) or 4-hour (proliferation) incubation in gentamycin-containing (25 µg/ml) medium before the cells were lysed with distilled water. Adhesion was defined as the total number of HeLa cell-associated bacteria and is shown as the percentage of input. Invasion and proliferation was defined as the total number of intracellular bacteria in HeLa cells. The mean ± SD of results from three independent experiments are shown. Statistical analysis was performed using student t-test. *, P < 0.01. For fluorescence microscopy in (C), GFP-expressing plasmid was electroporated into the Shigella strains, and adhesion assay was performed as described above, after washing, the cells were fixed in 3% paraformaldehyde/PBS and mounted with Anti-Fade solution (Invitrogen) containing DAPI onto glass slides. Shigella strains are in green and nuclei are blue. (D) SEM analysis of enhanced adhesion of ΔrfaC S.flexneri strain to HeLa cells. Bars in both images represent 30 µm. (E) Levels of LDH in HeLa cell culture supernatant, 3 h post-infection with S. flexneri. The mean ± SD of results from three independent experiments are shown. Statistical analysis was performed using student t-test. *, P < 0.01.

In vitro adhesiveness and thus invasiveness is promoted in biofilm-forming rfaC-deleted Shigella flexneri strain independent of activation of type III secretion system

Since biofilm formation is often associated with bacterial virulence, we set out to evaluate the pathogenic activities of the biofilm-forming LPS mutants using the gentamicin protection assay. The Δwzy and ΔwaaL strains showed almost identical host invasion efficiency to the WT strain. By contrast, the ΔrfaC strain showed a much stronger invasiveness and intracellular proliferation than wild type (Fig. 3A). Further dissection of time-dependent infection process revealed that the improved invasiveness came from enhanced adhesion during the initial host cell-Shigella contact (within 10 min), which was further confirmed by fluorescence microscopy and scanning electron microscopy (Figs. 3B–3D). To be specific, ΔrfaC mutant exhibited pronounced adhesiveness even when the type III secretion system (T3SS), responsible for bacterial invasion and virulence, was inactivated transcriptionally at low growth temperature (28–30°C). Of note, the invasion of low temperature-cultured bacteria was totally abolished as expected since no intracellular bacteria were recovered. To verify the cell-lysing ability of ΔrfaC mutant, we assayed the level of LDH in the cell culture medium 90 min post-infection. LDH is a stable cytoplasmic enzyme that is only released on loss of membrane integrity. The LDH level in the supernatant of ΔrfaC-infected HeLa cells was more than twice that of the M90TS-infected cell supernatant (Fig. 3E).

rfaC-deleted strain showed different actin-based motility from other LPS-truncated strains

To examine the distribution of IcsA in the LPS-truncated mutants, intracellular F-actin of the infected host cells were visualized by fluorescent phalloidin, which binds to F-actin of mammalian cells. Although F-actins could be recruited by all the Shigella strains except the icsA-deleted strain (Fig. 4A and Fig. S3), their cellular distribution varies between WT and LPS mutants. Typical “actin comets” and long protrusion were observed with the wild type strain (Fig. 4A), indicating a normal polar distribution of IcsA, which was further confirmed by the competence to form regular plaques on HeLa cell monolayer in a plaque assay (Fig. 4B). By contrast, the Δwzy and ΔwaaL mutants assembled F-actins all around the bacteria cell (Fig. S3), implying a circumferential distribution of IcsA, which caused aberrant actin-based motility (ABM) and thus failure to form plaques on HeLa cells monolayer (Fig. 4B). The ΔrfaC bacteria also recruited F-actin in a circumferential pattern as other LPS-truncated strains do. Nevertheless, a small amount of bacteria could assemble short and curly “actin-tail” behind the bacteria (as arrows indicated in Fig. 4A). Although ΔrfaC mutant could not form classical plaques on HeLa cell monolayer in 3 days, the integrity of the monolayer were destroyed due to massive cell detachment (Fig. 4B).

Figure 4 Actin-based motility of Shigella flexneri LPS mutants in comparison with the parental Sf301 strain.

(A) Fluorescence microscopy analysis of F-actin in HeLa infected by Sf301 and its ΔrfaC mutant strain. Invasion assay was performed as previously described. F-actin is stained by TRITC-phalloidin (red), Shigella bacteria are green and nuclei are blue (DAPI). White arrows in the WT panel indicate the long protrusion with actin comets assembled by Sf301. White arrows in the (triangle)rfaC panel indicate the short actin-tail formed by the ΔrfaC mutant bacteria. (B) Plaque formation of Sf301 and its LPS mutants on HeLa cell monolayer. Plaque assays were performed as described in Methods and Materials.

Truncation of LPS rendered bacteria susceptible in vitro and in vivo

LPS, especially long-chain LPS of gram-negative bacteria, provides protection for the bacteria against unfavorable environment. While loss of LPS may contribute to biofilm formation and pathogenic activities such as adhesion and invasion, the adverse effects associated with LPS deletion must also be taken into consideration to accurately gauge the overall influences on bacterial fitness. To do this, we measured the susceptibility of the LPS mutants to distilled water (to mimic the low osmotic shock in environment) and 10% pooled human serum (to mimic the adverse environment in vivo). While WT bacteria well survived the low osmotic shock (water) during the experiment, all three LPS mutants showed severely compromised viabilities after two hours in water. The ΔrfaC mutant in particular, even begun to exhibit significantly reduced viability within 30 min (Fig. 5A). Human serum showed much stronger killing activity than water toward Δwzy, ΔwaaL and ΔrfaC strains (Table 3). It is interesting to note that although most of the bacteria were killed by the serum, a small number of colonies were recovered for the ΔrfaC strain versus no colony for Δwzy and ΔwaaL strains (Fig. 5B). Whether this improved survivability of the ΔrfaC strain against serum is related to its biofilm-forming ability warrants further investigation.

Figure 5 In vitro susceptibility to water (A) and in vivo colonization ability (B) of Shigella flexneri LPS mutants in comparison with the parental Sf301 strain.

(A) Bacteria from mid- exponential phase were collected and resuspended into distilled water at the density of 107 CFU/ml, bacterial viability was evaluated as output CFU/input CFU X 100% at indicated time intervals. The mean ± SD of results from three independent experiments are shown. Statistical analysis was performed using student t-test. *, P < 0.01, against bacterial survival level of sf301 wild type. (B) To assess the colonization ability of the Shigella strains, 108 CFU were inoculated into the rectum of guinea pigs, and bacterial load in the feces was determined by the number of colonies on MacConkey Agar as described by Methods and Materials. The mean ± SD of results from three individual animals are shown.

Table 3 Bacterial viability of the strains after two-hour serum killing.

Treatment	Heated inactivated serum (%)	Pooled human serum (%)	
WT	90.6 ± 13.5	63.4 ± 6.4	
Δwzy	85.5 ± 7.6	0	
ΔwaaL	111.1 ± 11.4	0	
ΔrfaC	89.9 ± 10.5	0.0023 ± 0.0004	

Given the dramatic virulence-boosting phenotype of the ΔrfaC mutant in vitro, we next examined the in vivo effects of LPS mutants on virulence using two independent animal models. The Sereny test (guinea pig keratoconjunctivitis) indicated that Δwzy, ΔwaaL and Δrfac strains failed to cause any manifestation of conjunctivitis during a period of 2 weeks’ infection. Wild type, on the other hand, provoked a typical keratoconjunctivitis 72 h post-infection (Table 4). To test the colonizing ability of the “super-adhesive” ΔrfaC mutant in the intestine, guniea pigs were inoculated with 108 CFU of Shigella via intrarectal route. Feces analysis revealed that Δwzy, ΔwaaL and ΔrfaC failed to colonize the intestines of guinea pigs, and bacteria were evacuated in three days without any exhibited symptoms. Nevertheless, animals inoculated with wild type stably excreted bacteria in 18 tested days, albeit without observed symptoms either (Fig. 5B).

Table 4 Scores of Sereny test.

	WT	Δwzy	ΔwaaL	ΔrfaC	
	#1	#2	#3	#1	#2	#3	#1	#2	#3	#1	#2	#3	
day1	2	2	2	0	0	0	0	0	0	0	0	0	
day2	3	3	3	0	0	0	0	0	0	0	0	0	
day3	3	3	3	0	0	0	0	0	0	0	0	0	
day4	3	3	3	0	0	0	0	0	0	0	0	0	
day5	3	3	3	0	0	0	0	0	0	0	0	0	
day6	3	3	3	0	0	0	0	0	0	0	0	0	
day7	3	3	3	0	0	0	0	0	0	0	0	0	

Discussion

In the present study, we described a new and unusual phenotype of Shigella flexneri that is characteristic of enhanced biofilm formation. Deficiency in Hep synthesis in LPS due to deletion of rfaC gene resulted in “deep-rough” LPS with an incomplete inner core containing only the Kdo moieties. Although various LPS truncations have been studied for their effects on the biological activities of Shigella, the deep-rough mutant has never been tested. In this work, we found that the deep-rough Shigella (ΔrfaC) exhibited significantly enhanced biofilm-forming ability and strong adhesiveness toward host cells, which is in sharp contrast to the WT strain (Brotcke Zumsteg et al., 2014; Kline et al., 2009; Mahmoud et al., 2016; Schroeder & Hilbi, 2008). Besides, this deep-rough mutant could efficiently proliferate in the host cells and cause extensive cell lysis in few hours after infection. However, the pronounced in vitro virulence observed with this mutant was not recapitulated in vivo, most likely due to its increased vulnerability to environmental stress caused by LPS loss.

The biofilm of ΔrfaC Shigella showed a typical mesh-like structure and its formation is dependent on extracellular DNA as proven by DNase I treatment experiment. Previous studies have shown that LPS truncations in E. coli and Porphyromonas gingivalis also promoted biofilm formation (Nakao et al., 2012; Nakao, Senpuku & Watanabe, 2006), suggesting that the correlation between LPS loss and biofilm-forming capacity might be a common feature in gram-negative bacteria. Bacterial surface hydrophobicity is a major determinant for biofilm formation (Donlan, 2002; Mitzel et al., 2016). LPS truncation reduced the hydrophilic sugar moieties from bacteria surface and increased the exposure of the hydrophopic outer membrane lipid layer, thus enhancing the hydrophobicity on bacteria surface and facilitating biofilm formation.

WT Shigella flexneri adheres to host cells much less efficiently in vitro than other enterobacteria due to the lack of general adhesion apparatus like fimbriae (Edwards & Puente, 1998; Pizarro-Cerdá & Cossart, 2006; Snellings, Tall & Venkatesan, 1997). To our surprise, the ΔrfaC mutant exhibited extraordinary adhesion to host cells and manifested normal invasion and proliferation capabilities as well. The molecular mechanism of the hyperadhesiveness of ΔrfaC mutant remains unclear. Interestingly, we noticed that the ΔrfaC Shigella tended to form clusters on abiotic surface or host cells. Whether this cell–cell clustering is also a result of the improved hydrophobicity caused by complete LPS shedding requires further investigation. Nevertheless, it is conceivable that the clustered Shigella may adhere to host cells as one entity, in which numerous weak contacts contributed by each individual bacterium are pooled to improve the overall avidity of the bacterial cluster toward the host cell.

Previous studies have shown that the long-chain LPS was essential for maintaining polar distribution of IcsA (Doyle et al., 2015; Morona, Daniels & Den Bosch, 2003; Sandlin et al., 1995). Indeed, the Δwzy and ΔwaaL mutants with shortened LPS displayed circumferential distribution of IcsA. Interestingly, a small amount of ΔrfaC bacteria could assemble atypical, short and curly actin-tail, implying the heterogeneous distribution of IcsA in this new mutant. Results from plaque assay showed that ΔrfaC bacteria failed to form typical plaque on HeLa monolayer as the WT does, although it did caused mass host cell death eventually. This could be caused by a limited number of correctly assembled ABM in function or by premature killing of the host cell before the actin protrusions could reach adjacent cells due to much enhanced intracellular proliferation. How the ΔrfaC mutant retains certain level of functional IcsA warrants further investigation.

Although the ΔrfaC mutant showed a significantly boosted virulence in vitro, it failed to colonize in vivo. Susceptibility test against low osmotic shock or serum revealed that this mutant is highly sensitive to environmental stress. Clearly, the virulence benefit gained from biofilm formation and/or improved adhesion is not enough to compensate for the loss of fitness on survivability in vivo.

In conclusion, this study characterized a new Shigella flexneri mutant with deficiency in Hep synthesis of LPS. This mutant was capable of forming biofilm on abiotic surface and manifested extraordinary adhesiveness to host cells. Our study established a clear positive relationship between increased surface hydrophobicity and adhesiveness of Shigella flexneri. Although it is not realistic for a pathogenic bacteria to shed off LPS to gain hydrophobicity in vivo, our study raised a possibility that the host adhesiveness of Shigella may be modulated by altering the hydrophobicity of the bacteria.

Supplemental Information

Figure S1 PCR analysis of constructed S.flexneri LPS mutants in this study by agarose electrophoresis

PCR was performed using the primers listed in Table 2. DNA ladders were shown in Lane M. (A) Δwzy mutant strain. Lane 1, wild type (1238 bp); Lane 2, Δwzy: kana (1600 bp); Lane 3, 4, Δwzy (100 bp); (A) ΔwaaL mutant strain. Lane 1, wild type (1238 bp); Lane 2, ΔwaaL: kana (1600 bp); Lane 3, 4, ΔwaaL (100 bp); (A) ΔrfaC mutant strain. Lane 1, wild type (1238 bp); Lane 2, ΔrfaC: kana (1600 bp); Lane 3, 4, ΔrfaC (100 bp).

Click here for additional data file.

Figure S2 Influence of 48-hour DNaseI treatmenton viability of ΔrfaC strains

Click here for additional data file.

Figure S3 Fluorescencemicroscopy analysis of F-actin in HeLa infected by Sf301 and its ΔicsA, Δwzy and ΔwaaL mutant strains

Invasion assay was performed as previously described. F-actin is stained by TRITC-phalloidin (red), Shigella bacteria are green and nuclei are blue (DAPI).

Click here for additional data file.

We thank Dr. J Yu from Strathclyde Institute of Pharmacy and BioScience, University of Strathclyde (Glasgow, UK) for providing the Shigella flexneri 2a strain Sf301 and plasmids for λ Red recombination system. We thank P Wang from the Laboratory of Electron Microscopy of School of Medicine and J Yan from the Center for Biomedical Engineering and Regenerative Medicine of Frontier Institute of Science and Technology, Xi’an Jiaotong University, for their technical assistance with microscopic analysis.

Additional Information and Declarations

Competing Interests

Author Contributions

Animal Ethics

Data Availability

The authors declare there are no competing interests.

Dan Xu conceived and designed the experiments, performed the experiments, analyzed the data, contributed reagents/materials/analysis tools, wrote the paper, prepared figures and/or tables, reviewed drafts of the paper.

Wei Zhang performed the experiments, analyzed the data, reviewed drafts of the paper.

Bing Zhang and Chongbing Liao performed the experiments, reviewed drafts of the paper.

Yongping Shao conceived and designed the experiments, analyzed the data, contributed reagents/materials/analysis tools, wrote the paper, prepared figures and/or tables, reviewed drafts of the paper.

The following information was supplied relating to ethical approvals (i.e., approving body and any reference numbers):

The protocol has been approved by the Animal Research Ethical Committee of School of Life Science and Technology the Xi’an Jiaotong University. Approval No. 201411.

The following information was supplied regarding data availability:

The raw data has been supplied as Supplemental Dataset.

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
