# Peer review of "Characterization of a biofilm-forming Shigella flexneri phenotype due to deficiency in Hep biosynthesis"

_PeerJ, doi:10.7717/peerj.2178_

## Round 0.1 · original submission · Minor Revisions

As you can read from the report, the manuscript has received favorable comments from all three reviewers. They also raise some issues that I ask you to address in a rebuttal and appropriate revision.

Reviewer 1 ·

Basic reporting

Xu, Shao and colleagues report a new Shigella flexneri mutant with deficiency in Hep
synthesis of LPS. This mutant was capable of forming biofilm on abiotic surface and manifested extraordinary adhesiveness to host cells. They established a clear positive relationship between increased surface hydrophobicity and adhesiveness of Shigella flexneri. The article is well written and the structure of the article is consistent with the journal requirements. The data has been made available in accordance with PeerJ guidelines. All the figures are well presented.

Experimental design

This submission is within the scope of the journal, and the method in this paper is well presented and has potential merit in expanding to other field.

Validity of the findings

The work has been carefully done. And the interpretation is well balanced and supported by the data, and conclusions are valid.

Additional comments

Overall, this is a technically sound study and is written with clarity. I strongly support this manuscript to be published on Peerj without further revision.

Reviewer 2 ·

Basic reporting

The authors systematically characterized biofilm-forming Shigella flexneri phenotype by knock-out of wzy, waaL and rfaC genes in Hep biosynthesis. The authors firstly verified the genotypes and further used multiple assays to understand how the deficiency in biosynthesis affect the biofilm-forming phenotype. I think this study is well-performed with quality data.

Experimental design

The authors mainly looked into how LPS mutants behave in terms of bio-filming phenotype. The authors should provide more detailed info about the rationale that the key role of LPS biosynthesis related to bio-filming. Are wzy, waaL and rfaC genes most important for LPS biosynthesis? In Fig. 2A, the deletion of rfaC mainly positively contributed to biofilm formation, but not the other two genes. Are there any possible explanations regarding this as wzy and waaL are also involved in LPS biosynthesis.

In Page 15, line 229, what is 'TTSS'?

Validity of the findings

Although this study suggests a relationship between increased surface hydrophobicity and adhesiveness of Shigella flexneri, it is still not clear enough what is the role of LPS involved in such invasiveness and virulence.

Additional comments

It is well-known that LPS plays a key role in structural integrity of the bacteria and protects the membrane from certain kinds of chemical attack. The authors found rfaC-deleted strain showed different actin-based motility from other LPS-truncated strains. Based on this study, it looks inner core is even more important in adhesiveness and thus invasiveness for the bacteria compared to outer core. The authors should further clarify why ΔrfaC mutant showed enhanced virulence in vitro, but is very poor to colonize in vivo. In addition, it is not very clear that what is the relationship between LPS and cell surface hydrophobicity.

Reviewer 3 ·

Basic reporting

Authors stuied biofilm forming potentials and pathological behaviors of various LPS-truncated Shigella flexneri strains, and found a anti-correlation between biofilm forming capacity of Shigella and the LPS chain length in general. The results are interesting. The manuscript is well written.

Experimental design

All experiments were properly designed. Multiple techniques were utilized in this study.

Validity of the findings

The data presented in this manuscript is solid. All proper control experiments were performed.

Additional comments

Minor issue: The detailed method for statistic used in this manuscript is missing.

---

## Round 0.2 · accepted · Accept

We would like to accept your manuscript for publication.